# Query2Triple: Unified Query Encoding for Answering Diverse Complex Queries over Knowledge Graphs

**Yao Xu[1,2], Shizhu He[1,2,*], Cunguang Wang[3], Li Cai[3], Kang Liu[1,2,4], Jun Zhao[1,2]**

[1] The Laboratory of Cognition and Decision Intelligence for Complex Systems,
Institute of Automation, Chinese Academy of Sciences, Beijing, China
[2] School of Artificial Intelligence, University of Chinese Academy of Sciences, Beijing, China
[3] Meituan, Beijing, China
[4] Shanghai Artificial Intelligence Laboratory, Shanghai, China
{yao.xu, shizhu.he, kliu, jzhao}@nlpr.ia.ac.cn, {wangcunguang, caili03}@meituan.com

## Abstract

Complex Query Answering (CQA) is a challenge task of Knowledge Graph (KG). Due to the incompleteness of KGs, query embedding (QE) methods have been proposed to encode queries and entities into the same embedding space, and treat logical operators as neural set operators to obtain answers. However, these methods train KG embeddings and neural set operators concurrently on both simple (one-hop) and complex (multi-hop and logical) queries, which causes performance degradation on simple queries and low training efficiency. In this paper, we propose Query to Triple (Q2T), a novel approach that decouples the training for simple and complex queries. Q2T divides the training into two stages: (1) Pre-training a neural link predictor on simple queries to predict tail entities based on the head entity and relation. (2) Training a query encoder on complex queries to encode diverse complex queries into a unified triple form that can be efficiently solved by the pretrained neural link predictor. Our proposed Q2T is not only efficient to train, but also modular, thus easily adaptable to various neural link predictors that have been studied well. Extensive experiments demonstrate that, even without explicit modeling for neural set operators, Q2T still achieves state-of-the-art performance on diverse complex queries over three public benchmarks.

## 1 Introduction

Knowledge Graphs (KGs) organize world knowledge as inter-linked triples which describe entities and their relationships in symbolic form (Ji et al., 2020). Complex query answering (CQA), a knowledge reasoning task over KGs, has been proposed in recent years (Wang et al., 2021). Compared with simple link prediction, which involves predicting a missing entity in a factual triple (Rossi et al., 2021), CQA is more challenging as it requires first-order

---

[*] Corresponding Author

Figure 1: An example, along with its corresponding interpretation and query graph, in CQA.

logic (FOL) operators such as existential quantification ($\exists$), conjunction ($\wedge$), disjunction ($\vee$), and negation ($\neg$) to be performed on entities, relations and triples, as shown in Figure 1. As KGs usually suffer from incompleteness (West et al., 2014), traditional query methods such as SPARQL (Prudhommeaux et al., 2013) cannot handle CQA well on real-world KGs such as Freebase (Bollacker et al., 2008) and NELL (Carlson et al., 2010).

Recently, an alternative method called Query Embedding (QE) (Hamilton et al., 2018; Ren et al., 2020) has been proposed to complete the missing edges and answer the complex query simultaneously (Wang et al., 2023b). The idea of these methods is to encode queries and entities into the same embedding space, and treats logical operators as neural set operators (e.g. relation projection as set projection, queries conjunction as sets intersection, query negation and set complement) to answer queries. QE methods embed a query by iteratively performing neural set operators according to the topological order of nodes in the query graph, and then obtain answers based on the similarity scores between the query embedding and all entity embeddings. This process is shown in Figure 3 (B).

Despite QE-based methods achieving good performance on CQA, they still exhibit the following drawbacks: (1) **Cannot perform well on both sim-**

**ple and complex queries simultaneously** (simple queries refer to one-hop queries, complex queries encompass multi-hop and logical queries). One potential reason from the model perspective is that, QE-based models model the projection operator with complex neural networks to perform well on complex queries, however, these complex neural networks are incapable of effectively managing inherent relational properties (Wang et al., 2023b), such as symmetry, asymmetry, inversion, composition, which are sufficiently studied in KG completion tasks and addressed by Knowledge Graph Embedding (KGE) (Wang et al., 2017), so KGE models can easily outperform QE-based methods on simple queries. Another potential reason from the training perspective is that, the training for simple queries and complex queries are coupled in these methods, as demonstrated in Figure 2 (A). Therefore, the entity and relation embeddings inevitably incorporate neural set operators information that is irrelevant for answering simple queries, leading to a decrease in performance. (2) **Low training efficiency and not modular**. From the model perspective, as the KGE is coupled with the neural set operators in QE-based methods, as shown in Figure 2 (A), these methods usually train KGE and neural set operators from scratch in each round of training, which is inefficient. Furthermore, due to this coupling, these methods cannot directly exploit other high-performance KGE models such as ComplEx (Trouillon et al., 2016). (3) **Suffer from error cascading**. These methods have to calculate all intermediate node representations step-by-step, as demonstrated in Figure 3 (B), so errors will be cascaded along the path (Guu et al., 2015), which affects the quality of answers.

Aiming to handle the above drawbacks, in this paper, we propose Query to Triple (Q2T), a novel approach that not only decouples the training of complex queries and simple queries from the training perspective, but also decouples KGE and the query encoder from the model perspective. Q2T divides the training into two stage: pre-training KGE for simple queries and training a unified query encoder for diverse complex queries, as shown in Figure 2 (B). In the first pre-training stage, we only train a KGE model as a neural link predictor on simple queries, such as DistMult (Yang et al., 2015) and ComplEx (Trouillon et al., 2016). Motivated by the prompt (Sun et al., 2022b; Liu et al., 2023) technology in Natural Languages Process (NLP),

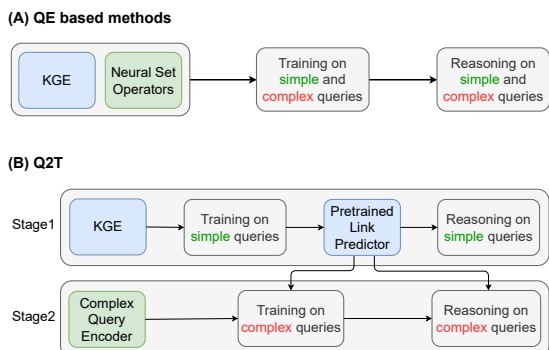

Figure 2: The different training strategies used in QE-based methods and Q2T. KGE denotes entity and relation embeddings. (A) QE-based methods train KGE and neural set operators on both simple and complex queries. (B) Q2T train KGE and Query Encoder on simple and complex queries respectively.

which aims to fully reuse the pretrained model, in the second stage, Q2T utilizes a trainable encoder to transform any complex query graph $g$ into a unified triple form $(\boldsymbol{g_h}, \boldsymbol{g_r}, t_?)$, where $\boldsymbol{g_h}, \boldsymbol{g_r}$ are representations of the whole query graph generated by the encoder, $t_?$ are tail entities to be predicted. Then, we can obtain probability list of all entities by feeding these two representations and all pretrained entity embeddings to the pretrained neural link predictor. That is, we transformer any complex query to a unified triple form which can be handled well by neural link predictors. In other words, as shown in Figure 3 (A), **Q2T treats CQA as a special link prediction task where it learns how to generate appropriate inputs for the pretrained neural link predictor** instead of reasoning in the embedding space step-by-step.

The advantages of Q2T are as follows: (1) Good performance on both simple and complex queries, because Q2T can not only make full use of the well-studies KGE models but also decouple the the training of simple queries and complex queries from the training perspective. (2) High training efficiency and modular, because from the model perspective, Q2T decouples the KGE and the query encoder, so it only trains the encoder in each round of training (pretrained KGE is frozen) and can also fully utilize various advanced KGE models. (3) Simple and accurate, as Q2T utilizes an end-to-end reasoning mode, which not only eliminates the need to model complex neural set operators, but also avoids error cascading.

We conducted experiments on widely-used benchmark such as FB15k (Bordes et al., 2013),

FB15k-237 (Toutanova and Chen, 2015), and NELL995 (Xiong et al., 2017). The experimental results demonstrate that, even without explicit modeling for neural set operators, Q2T still achieves state-of-the-art performance on diverse complex queries. The source codes and data can be found at `https://github.com/YaooXu/Q2T`.

## 2 Related work

**Knowledge Graph Embedding (KGE)**. KGE embeds components of a KG including entities and relations into continuous vector spaces, so as to predict unseen relational triples while preserving the inherent structure of the KG (Wang et al., 2017). According to the different scoring functions, KGE can be roughly categorized into two types: translational distance models (Bordes et al., 2013; Wang et al., 2014) and semantic matching models (Yang et al., 2015; Trouillon et al., 2016). All these models can be used in Q2T as neural link predictors.

**Prompt-based Learning**. In recent years, a new training paradigm called "pre-train, prompt, and predict" has achieved success in both NLP (Liu et al., 2023) and graphs (Sun et al., 2022a). Aiming to effectively reuse the pretrained model, this paradigm targets reformulating the downstream task looks similar to pre-training task (Liu et al., 2023). Based on the same idea, in Q2T, we treat simple and complex queries as the pre-training and downstream tasks respectively. The difference is that we use a trainable encoder to encode complex queries to the same triple form as simple queries.

**Neural complex query answering**. Neural complex query answering can be divided into two categories: with/without pretrained KGE. Query Embedding (QE) based models, without pretrained KGE, encode queries and entities into the same embedding space, then obtain answers by step-by-step reasoning, as shown in Figure 3 (B). In QE-based models, many methods have been proposed to represent entity sets, such as geometric shapes (Ren et al., 2020; Bai et al., 2022), probability distribution (Ren and Leskovec, 2020) and bounded histogram on uniform grids (Wang et al., 2023a). Neural networks like multi-layer perceptron and transformers (Vaswani et al., 2017) are used to model set operations. They are trained for simple and complex queries concurrently, so they suffer drawbacks mentioned before.

Models with pretrained KGE are as follows: (1) CQD (Arakelyan et al., 2021) uses the pretrained neural link predictor as the one-hop reasoner and T-norms as logic operators to obtain the continuous truth value of an EPFO query. Then the embeddings are optimized to maximize the continuous truth value. However, this method is time consuming and cannot handle negation queries well (Wang et al., 2023b). (2) LMPNN (Wang et al., 2023b), which is related to our work, also utilizes pretrained KGE to conduct one-hop inferences on atomic formulas, the generated results are regarded as the messages passed in Graph Neural Network (GNN) (Wu et al., 2020). However, in the LMPNN, KGE and GNN are coupled, which means the KGE function is invoked during every message propagation on each edge in the inference. Besides, LMPNN has to design of distinct logical message propagation functions for different types of score functions in KGE, as well as for different types of regularization optimization. Compared to LMPNN, Q2T only requires a single invocation of the KGE score function for any type of complex query. Therefore, Q2T is better suited for leveraging KGE with higher complexity and larger parameters. Additionally, Q2T can be employed with any KGE without the need for additional code modifications, so it is more modular.

SQE (Bai et al., 2023) is also related to our work, it uses a search-based algorithm to linearize the computational graph to a sequence of tokens and then uses a sequence encoder to compute its vector representation. Another related work is KgTransformer (Liu et al., 2022). It also uses the Transformer to encode queries, but it outputs the final node embedding directly. To improve transferability and generalizability, KgTransformer introduces a two-stage masked pre-training strategy, which requires more pre-training data and time. However, Q2T can outperform KgTransformer easily with much less parameters and training time.

## 3 Preliminary

In this section, we formally introduce knowledge graph (KG). Then, we use symbols of KG to define EFO-1 query and its corresponding Disjunctive Normal Form (DNF). Finally, we introduce neural link predictors.

### 3.1 Knowledge Graph and EFO-1 Queries

We represent a KG as a set of factual triples, i.e., $G = \{(h, r, t) \in \mathcal{V} \times \mathcal{R} \times \mathcal{V}\}$, where $h, r \in \mathcal{V}$ denote the head and tail entity, $r \in \mathcal{R}$ represents

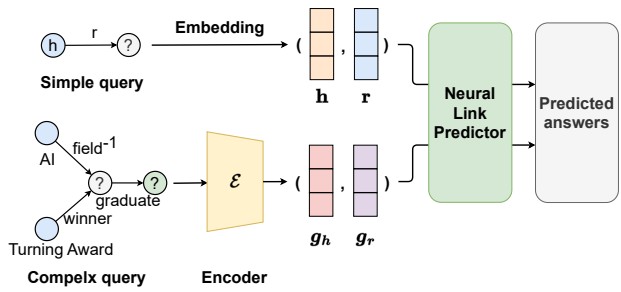
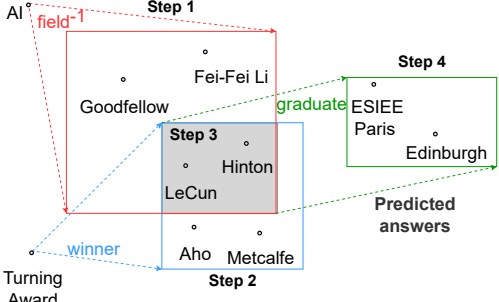

**(A) The end-to-end reasoning mode in Q2T**

**(B) The step-by-step reasoning mode in QE**

Figure 3: The different reasoning mode for the query *"Where did the person who won the Turing Award in AI graduate from?"*. (A) Q2T adopts end-to-end reasoning mode: encoding the query graph to obtain two representation $g_h, g_r$, then the pretrained neural link predictor takes $g_h, g_r$ as inputs and output predicted tail entities, which is similar to answering simple queries. (B) QE-based methods utilize step-by-step reasoning mode: iteratively apply the neural set operators to get answers. Step 1 and Step 2 denote applying set projection from *AI* and *Turning Award* respectively. Step 3 denotes applying set intersection to the results of Step 1 and Step 2. Finally, Step 4 denotes applying set projection to the result obtained in Step 3.

the relation. We make $r(h, r) = 1$ if and only if there is a directed relation $r$ between $h$ and $t$.

Following previous work (Ren and Leskovec, 2020), we define the Existential First Order queries with a single free variable (EFO-1), and an arbitrary EFO-1 logical formula can be converted to the Disjunctive Normal Form (DNF) as follows:

$$q[V_?] = V_?. \exists V_1, ..., \exists V_n : c_1 \vee ... \vee c_m \quad (1)$$

where $V_?$ is the only free variable and $V_i, 1 \leq i \leq n$ are $n$ existential variables, each $c_i, 1 \leq i \leq m$ represents a conjunctive clause like $c_i = e_{i1} \wedge e_{i2} \wedge ... \wedge e_{ik_i}$, and each $e_{ij}, 1 \leq j \leq k_i$ indicates an atomic formula or its negation, i.e., $e_{ij} = r(a, b)$ or $\neg r(a, b)$, where $r \in \mathcal{R}, a, b$ can be either a constant $e \in \mathcal{V}$ or a variable $V$.

### 3.2 Neural Link Predictor

KGE is a kind of neural link predictor, for simplicity, neural link predictors mentioned in this article refer to KGE models. Given the head entity embedding $\mathbf{h}$, relation embedding $\mathbf{r}$, and tail entity embedding $\mathbf{t}$, KGE can compute the score $\phi(\mathbf{h}, \mathbf{r}, \mathbf{t})$ of the triple $(h, r, t)$, where $\phi(\mathbf{h}, \mathbf{r}, \mathbf{t})$ indicates the likelihood that entities $h$ and $t$ hold the relation $r$.

## 4 Q2T: Transforming Query to Triple

There are two main components in Q2T: (1) The Neural Link Predictor. (2) The Query Encoder. This section begins with an introduction to the pretraining of neural link predictors. Subsequently, we present QueryGraphormer, a Transformer-based

model that incorporates structural information, as a query encoder. Q2T utilizes the query encoder to encode complex queries, producing representations that are used as inputs in neural link predictors, as depicted in Figure 4.

### 4.1 The Pretrained Neural Link Predictor

As answers of complex queries are generated by the pretrained neural link predictor finally, its performance directly affects the final performance.

Aiming to get neural link predictors with good performance, the training object we use contains not only terms for predicting the tail entity of a given triple, but also a term for predicting the relation type, which has been proved to significantly improve entity ranking (Chen et al., 2021). The training object is defined as follows:

$$\arg\max_{\theta \in \Theta} \sum_{(h,r,t) \in \mathcal{G}} [log P_\theta(t|h, r) + \lambda log P_\theta(r|h, t)]$$

$$log P_\theta(t|h, r) = \phi(\mathbf{h}, \mathbf{r}, \mathbf{t}) - log \sum_{t' \in \mathcal{E}} \exp\left[\phi(\mathbf{h}, \mathbf{r}, \mathbf{t}')\right]$$

$$log P_\theta(r|h, t) = \phi(\mathbf{h}, \mathbf{r}, \mathbf{t}) - log \sum_{r' \in \mathcal{R}} \exp\left[\phi(\mathbf{h}, \mathbf{r}', \mathbf{t})\right]$$

$$(2)$$

where $\theta \in \Theta$ are the model parameters, including entity and relation embeddings, $h, r, t$ denote head entity, relation and tail entity, $\mathbf{h}, \mathbf{r}, \mathbf{t}$ denote their corresponding embeddings under $\theta$, $\phi$ is a scoring function, $\lambda$ is a hyper-parameter determining the contribution of the relation prediction objective.

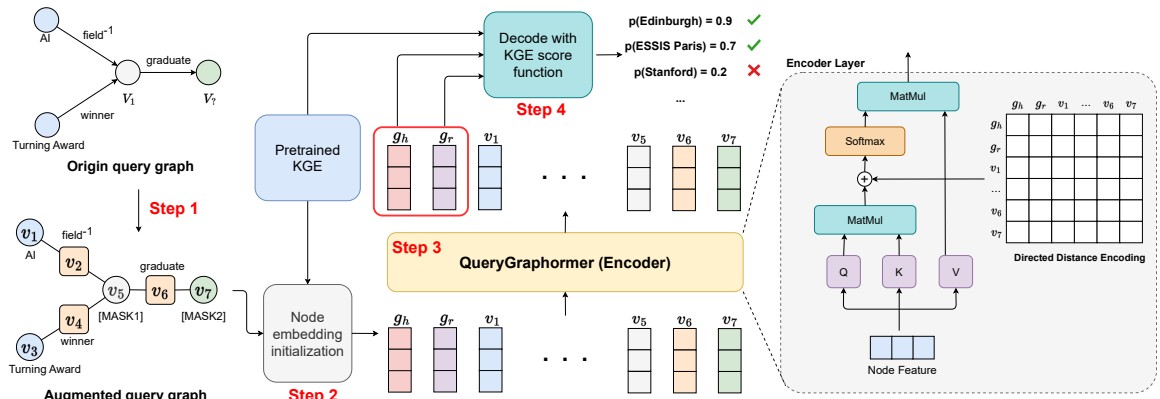

Figure 4: The framework of Q2T. The steps of answering complex queries in Q2T are: (1) Transforming the origin query graph to a augmented graph without edge attributes. (2) Flattening the augmented graph to a sequence, and initializing the sequence with the pretrained KGE. (3) Encoding the sequence representations with QueryGraphormer which considers the structural information of the graph. (4) Inputting the final representations of $g_h, g_r$, and other entity representations to the score function of the pretrained KGE to calculate the probability list of entities.

## 4.2 The QueryGraphormer

**Transformer for CQA**   With the great success of Transformer (Vaswani et al., 2017) in NLP, Transformer has also been proved to perform well for graph representation learning (Ying et al., 2021). KgTransformer (Liu et al., 2022) is the first work to apply Transformer on query graphs, which first turns relations into relation-nodes to transform the query graph into an augmented graph without edge attributes, then masks adjacency matrices (Hu et al., 2020) to self-attention. However, this limits receptive field of each node, which means each node can only obtain information from its neighbors.

Motivated by Graphormer (Ying et al., 2021), we propose QueryGraphormer, which uses the directed distance between any two nodes as a bias term in the attention mechanism, to solve these drawbacks. This helps the model effectively capture spatial dependencies within the augmented graph (obtained through the same transformation as KgTransformer).

The biggest difference between our proposed QueryGraphormer and KgTransformer is that outputs of QueryGraphormer are inputted to pretrained neural link predictors to get the final predicted answers, while KgTransformer outputs the final predicted answer embedding directly.

### 4.2.1 Input Representations Construction

In this part, we introduce how to construct input sequence representations for QueryGraphormer.

**Initial input representations.**   Similar to BERT (Devlin et al., 2019) model which introduces a spe-

cial token $[CLS]$ to represent the sentence-level feature, in QueryGraphormer, we add two special node: $[g_h]$ and $[g_r]$, which represent a virtual head node and a virtual relation node respectively. In the view of graph, $[g_h]$ and $[g_r]$ can be regraded as two virtual nodes connected to all nodes in the query graph. We attach $[g_h]$ and $[g_r]$ at the beginning of each sequence, then a query graph is flattened into a sequence, $S = [g_h, g_r, v_1, ..., v_n]$, where $n$ is the number of nodes in the augmented graph, $n_i, i \leq n$ is an entity node, relation node or MASK node. For each node, its input representation is a frozen embedding from the pretrained KGE if it is an entity node or relation node, or a trainable embedding otherwise. We denote the initial representation of $S$ as $\boldsymbol{S^0} \in \mathbb{R}^{m \times d_0}$, where $m = n + 2$, $d_0$ is the dimension of the pretrained KGE.

**Projection.**   After obtaining the initial representations of $S$, we use MLP to project features from high-dimensional to low-dimensional to reduce the parameter amount and calculation time of the QueryGraphormer, which is described as follows:

$$\boldsymbol{S^1} = MLP_{proj}(\boldsymbol{S^0}) \qquad (3)$$

where $\boldsymbol{S^1} \in \mathbb{R}^{m \times d_1}$ and $\boldsymbol{S^0} \in \mathbb{R}^{m \times d_0}$ are the inital/projected representations of $S$, and $d_1 < d_0$.

**Handle negation queries.**   Aiming to make the input representations of negation queries different from positive queries. We apply a linear transformation on node representation that represents a negation relation: $\boldsymbol{h_i^{neg}} = \boldsymbol{A h_i}$, where $\boldsymbol{A} \in \mathbb{R}^{d_1 \times d_1}$, $\boldsymbol{h_i} \in \mathbb{R}^{d_1}$ denotes the the $i$-th element (negation relation node) in $\boldsymbol{S^1}$.

### 4.2.2 Self-Attention with Distance Message

**Directed Distance Encoding.** Similar to the *Spatial Encoding* used in Graphormer (Ying et al., 2021), we employ a function $\phi(v_i, v_j) : V \times V \to \mathbb{R}$ which measures the spatial relation between $v_i$ and $v_j$ in the query graph $g$. Unlike Graphormer which uses the distance of the shortest path to be the function, in this paper, we use the directed distance of the shortest path, because the topological order is crucial in query graphs, that is, predecessors and successors of each node should be treated differently.

The directed distance of the shortest path is defined as follows:

$$\phi(v_i, v_j) = dir * spd(v_i, v_j) \quad (4)$$

$$dir = sgn(layer(v_i) - layer(v_j)) \quad (5)$$

where $spd(v_i, v_j)$ denotes the distance of the shortest path between $v_i$ and $v_j$, $layer(v_j)$ denotes the layer number of $v_j$ (obtained by topological sort), $sgn(x)$ is the sign function: $sgn(x) = 1$ if $x \geq 0$, $sgn(x) = 0$ if $x < 0$.

For each output value of $\phi$, we allocate a trainable scalar to be a bias term in the self-attention. Then the attention $\alpha_{ij}$ between $v_i$ and $v_j$ is computed in the following way (for simplicity, we only consider single-head self-attention here):

$$\alpha_{ij} = \frac{\exp a_{ij}}{\sum_{k=1}^{m} \exp a_{ik}} \quad (6)$$

$$a_{ij} = \frac{(h_i W^Q)(h_j W^K)^T}{\sqrt{d_1}} + b_{\psi(v_i, v_j)} \quad (7)$$

where $W^Q, W^K \in \mathbb{R}^{d_1 \times d_1}$ are Query and Key matrices, $b_{\psi(v_i, v_j)}$ is a learnable scalar indexed by $\psi(v_i, v_j)$. By doing this, each node in the query graph can not only aggregate the information of all other nodes, but also decide which nodes should pay more attention to based on distance messages.

The result of self-attention for $v_i$ is obtained as follows:

$$Attn_i = \sum_{j=1}^{n} \alpha_{ij}(h_j W^V) \quad (8)$$

where $W^V \in \mathbb{R}^{d_1 \times d_1}$ is Value matrix.

Then we adopt the classic Transformer encoder to get the output representations of layer $l$, which is described as follows:

$$h'^{(l)} = LN(MHA(h^{(l-1)}) + h^{(l-1)}) \quad (9)$$

$$h^{(l)} = LN(FFN(h'^{(l)}) + h'^{(l)}) \quad (10)$$

$$FFN(x) = act(x W_1 + b_1) W_2 + b_2 \quad (11)$$

where MHA denotes Multi-Head Attention described before, FFN denotes Feed-Forward blocks, LN denotes layer normalization, $W_1 \in \mathbb{R}^{d_1 \times d_2}, W_2 \in \mathbb{R}^{d_2 \times d_1}$, and $act$ is the activation function. In QueryGraphormer, we make $d_1 = d_2$.

### 4.2.3 Training Object of QueryGraphormer

Finally, we feed the representations of $g_h$ and $g_r$ in the last layer, denoted as $g_h, g_r \in \mathbb{R}^{d_1}$, into the pretrained neural link predictor to obtain predicted answers. Formally, given a query graph $g$, the score of each tail entity $t_i$ is calculated as follows:

$$s(t_i \,|\, g) = \phi(g_h{}', g_r{}', t_i) \quad (12)$$

$$g_h{}' = MLP_{rev}(g_h), \; g_r{}' = MLP_{rev}(g_r) \quad (13)$$

where $MLP_{rev}$ projects the representations from $d_1$ dimension back to $d_0$ dimension, $t_i \in \mathbb{R}^{d_0}$ is the pretrained embedding of entity $t_i$, $\phi$ is the score function of the pretrained KGE.

It is noteworthy that $t_i$ is from the pretrained entity embeddings, and they are not needed to updated in the training of QueryGraphormer, which effectively reduces the number of parameters needed to be updated.

Our training object is to maximize the log probabilities of correct answer $t$, the loss function (without label smoothing for simplicity) is defined as follows:

$$L = -(s(t \,|\, g) - log \sum_{i=1}^{K} \exp[s(t_i \,|\, g)]) \quad (14)$$

where $t$ and $t_i$ are the answer entity and negative entity (random sampling), respectively.

## 5 Experiments

In this section, we conduct experiments to demonstrate the effectiveness and efficiency of Q2T and the necessity of decoupling the training for simple and complex queries.

### 5.1 Experiment Setup

**Datasets.** To compare our results and previous works directly, we conduct our experiments on the widely used datasets generated by Ren and Leskovec. The logical queries in these datasets are generated from FB15k (Bordes et al., 2013), FB15k-237 (Toutanova and Chen, 2015), and NELL995 (Xiong et al., 2017). More details about these queries can be found in Appendix A.

| Dataset | Model | 1p | 2p | 3p | 2i | 3i | pi | ip | 2u | up | 2in | 3in | inp | pin | pni | $A_p$ | $A_n$ |
|---|---|---|---|---|---|---|---|---|---|---|---|---|---|---|---|---|---|
| **FB15k** | BetaE | 65.1 | 25.7 | 24.7 | 55.8 | 66.5 | 43.9 | 28.1 | 40.1 | 25.4 | 14.3 | 14.7 | 11.5 | 6.5 | 12.4 | 41.6 | 11.8 |
| | Q2P | 82.6 | 30.8 | 25.5 | 65.1 | 74.7 | 49.5 | 34.9 | 32.1 | 26.2 | 21.9 | 20.8 | 12.5 | 8.9 | 17.1 | 46.8 | 16.4 |
| | ConE | 73.3 | 33.8 | 29.2 | 64.4 | 73.7 | 50.9 | 35.7 | 55.7 | 31.4 | 17.9 | 18.7 | 12.5 | 9.8 | 15.1 | 49.8 | 14.8 |
| | (with pretrained KGE) | | | | | | | | | | | | | | | | |
| | CQD-CO | 89.2 | 25.6 | 13.6 | **77.4** | 78.3 | 44.2 | 33.2 | 41.7 | 22.1 | - | - | - | - | - | 46.9 | - |
| | LMPNN | 85.0 | 39.3 | 28.6 | 68.2 | 76.5 | 46.7 | 43.0 | 36.7 | 31.4 | **29.1** | **29.4** | 14.9 | **10.2** | 16.4 | 50.6 | **20.0** |
| | Q2T | **89.4** | **44.3** | **33.6** | 72.1 | **79.4** | **55.3** | **47.7** | **65.8** | **38.2** | 19.5 | 21.3 | **16.2** | **10.2** | **16.7** | **58.4** | 16.8 |
| **FB15k -237** | BetaE | 39.0 | 10.9 | 10.0 | 28.8 | 42.5 | 22.4 | 12.6 | 12.4 | 9.7 | 5.1 | 7.9 | 7.4 | 3.6 | 3.4 | 20.9 | 5.4 |
| | Q2P | 39.1 | 11.4 | 10.1 | 32.3 | 47.7 | 24.0 | 14.3 | 87.0 | 9.1 | 4.4 | 9.7 | 7.5 | 4.6 | 3.8 | 21.9 | 6.0 |
| | ConE | 41.8 | 12.8 | 11.0 | 32.6 | 47.3 | 25.5 | 14.0 | 14.5 | 10.8 | 5.4 | 8.6 | 7.8 | 4.0 | 3.6 | 23.4 | 5.9 |
| | FuzzQE | 42.2 | 13.3 | 10.2 | 33.0 | 47.3 | 26.2 | 18.9 | 15.6 | 10.8 | **9.7** | 12.6 | 7.8 | **5.8** | **6.6** | 24.2 | **8.5** |
| | (with pretrained KGE) | | | | | | | | | | | | | | | | |
| | CQD-CO | 46.7 | 9.6 | 6.2 | 31.2 | 40.6 | 23.6 | 16.0 | 14.5 | 8.2 | - | - | - | - | - | 21.9 | - |
| | LMPNN | 45.9 | 13.1 | 10.3 | 34.8 | 48.9 | 22.7 | 17.6 | 13.5 | 10.3 | 8.7 | **12.9** | 7.7 | 4.6 | 5.2 | 24.1 | 7.8 |
| | Q2T | **48.4** | **15.6** | **12.4** | **37.8** | **51.9** | **28.6** | **19.4** | **21.8** | **12.8** | 6.1 | 12.4 | **8.4** | 4.5 | 4.1 | **27.6** | 7.1 |
| **NELL** | BetaE | 53.0 | 13.0 | 11.4 | 37.6 | 47.5 | 24.1 | 14.3 | 12.2 | 8.6 | 5.1 | 7.8 | 10.0 | 3.1 | 3.5 | 24.6 | 5.9 |
| | Q2P | 56.5 | 15.2 | 12.5 | 35.8 | 48.7 | 22.6 | 16.1 | 11.1 | 10.4 | 5.1 | 7.4 | 10.2 | 3.3 | 3.4 | 25.5 | 6.0 |
| | ConE | 53.1 | 16.1 | 13.9 | 40.0 | 50.8 | 26.3 | 17.5 | 15.3 | 11.3 | 5.7 | 8.1 | 10.8 | 3.5 | 3.9 | 27.2 | 6.4 |
| | FuzzQE | 58.1 | 19.3 | 15.7 | 39.8 | 50.3 | 28.1 | 21.8 | 17.3 | 13.7 | 8.3 | 10.2 | 11.5 | **4.6** | **5.4** | 29.3 | **8.0** |
| | (with pretrained KGE) | | | | | | | | | | | | | | | | |
| | CQD-CO | 60.4 | 17.8 | 12.8 | 39.3 | 46.6 | 30.1 | 22.1 | 17.3 | 13.2 | - | - | - | - | - | 28.8 | - |
| | LMPNN | 60.6 | **22.1** | 17.5 | 40.1 | 50.3 | 28.4 | **24.9** | 17.2 | **15.7** | 8.5 | **10.8** | 12.2 | 3.9 | 4.8 | 30.7 | **8.0** |
| | Q2T | **61.1** | **22.1** | **18.0** | **41.1** | **52.3** | **30.3** | 23.4 | **19.5** | 15.3 | 5.8 | 7.5 | 11.3 | 3.7 | 4.3 | **31.5** | 6.5 |

Table 1: MRR results of different CQA models over three KGs. $A_P$ and $A_N$ denote the average score of EPFO queries (1p/2p/3p/2i/3i/pi/ip/2u/up) and queries with negation (2in/3in/inp/pin/pni), respectively. The boldface indicates the best result of each query type.

| | $A_p$ | $A_n$ |
|---|---|---|
| Q2T (no structrual information) | 30.2 | 5.6 |
| - Adjacency Matrices Masking | 30.5 | 6.4 |
| - Undirected Distance Encoding | 31.0 | **6.5** |
| - Directed Distance Encoding | **31.5** | **6.5** |

Table 2: MRR of models with different structural information encoding strategies on the NELL dataset.

**Evaluation.** The evaluation metrics follows the previous works (Ren and Leskovec, 2020), which aims to measure the ability of models to discover *hard* answers. That is, these answers cannot be found in the training KG by symbolic methods due to missing edges. Concretely, for each hard answer $v$ of a query $q$, we rank it against non-answer entities and calculate the Mean Reciprocal Rank (MRR) as evaluation metrics.

**Baselines.** We mainly compare our work with non symbolic-integrated CQA models for EFO-1 queries. The baselines can be divided into two types: (1) Without pretrained KGE, such as BetaE (Ren and Leskovec, 2020), ConE (Zhang et al., 2021), Q2P (Bai et al., 2022) and FuzzQE (Chen et al., 2022). (2) With pretrained KGE, such as CQD-CO (Arakelyan et al., 2021), and LMPNN (Wang et al., 2023b).

As some works (e.g. KgTransformer (Liu et al., 2022)) do not support negation queries, they use the

EPFO queries (queries without negation) generated by Ren et al., we also compare these works on the EPFO datasets in Appendix F. We also compare our work with symbolic-integrated CQA models in Appendix G.

## 5.2 Comparison with Baselines

Table 1 shows the MRR results of Q2T and baselines on EFO-1 queries over three benchmarks. The experiment settings can be found in Appendix B.

From the table, we can find that our Q2T can achieve the state-of-the-art performance on almost all positive queries over three datasets. Especially on FB15k and FB15k-237, our Q2T has a significant improvement on EPFO performance, achieving 15.4% and 14.9% relative improvement on $A_P$ respectively. Furthermore, compared to the step-by-step reasoning mode, the end-to-end reasoning mode can eliminate error cascading to some extent, which is reflected in the better performance on multi-hop queries (2p/3p). More details about the model parameters can be found in Appendix C.

This results show that, even without the explicit modeling for neural set operators, our Q2T can still perform well on logical queries, one potential explanation is as follows: **Each pretrained neural link predictor can be regarded as a neural knowledge base which record the structure**

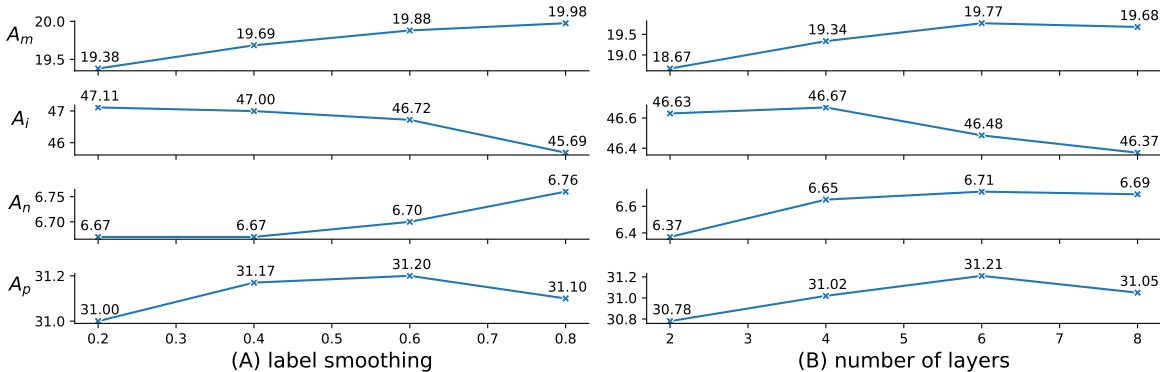

Figure 5: Hyperparameter analysis on NELL. $A_m, A_i, A_n, A_p$ denote the average MRR of multi-hop queries, intersection queries, negation queries, and all positive queries. Since the performance is too low when the label smoothing is 0, they are not shown in the figure (A).

**of KG, while learning to transformer complex queries into suitable inputs for the link predictor can be seen as an end-to-end process of retrieving answers from this neural knowledge base.**

Compared to BetaE, Q2P and ConE, Q2T's strong performance on both simple and complex queries underscores the critical importance of separating their training processes.

Although Q2T cannot outperform previous models on negation queries, it still gets competitive performance. One potential explanation for this lies in the fact that the neural link predictor is exclusively pretrained on positive triples. The distribution of answer entities for positive triples tends to be more concentrated, whereas queries containing negation operators result in a more dispersed answer entity distribution. For example, the "complement" operator can yield numerous candidate answers, as seen in queries like "Who is not the winner of the Turing Award." Consequently, pretrained KGE models struggle to generate suitable answer embeddings for queries with dispersed answer distributions.

We also test the performance of Q2T with different pretrained KGE on the FB15k-237 queries, and the results are shown in Appendix E.

### 5.3 Ablation Studies

In this section, we conduct further experiments to demonstrate the effectiveness of Directed Distance Encoding and investigate how hyperparameters affect the performance on each query type.

**Structure Encoding.** We compare our proposed Directed Distance Encoding to two commonly used strategies: Undirected Distance Encoding and Adjacency Matrices Masking, which are described in section 4.2. The results are reported in Table 2, it

can be found that models with structural information perform well consistently in CQA, indicating that structural information is necessary. Besides, Distance Encoding that leverages global receptive field is better than Adjacency Matrices Masking that uses local receptive field. Besides, compared to undirected distance, directed distance can further improve performance by introducing extra topological information that help each node to treat their predecessors and successors differently.

**Label smoothing.** Label smoothing is crucial in Q2T, model without label smoothing suffer from severe overfitting, resulting in bad performance on all queries. More details about label smoothing can be found in Appendix D. Figure 5 (A) shows how label smoothing influences performance, it can be observed that different types of queries prefer different label smoothing values, multi-hop and negation queries prefer high label smoothing value while intersection queries do the opposite, we think this difference is caused by the difference in the average number of answers, as shown in Appendix A. **More specifically, queries with more candidate answers prefer higher label smoothing value.**

**Number of layers.** To justify how the number of layers affects Q2T, we test the performance of models with a different number of layers on different query types. Figure 5 (B) demonstrates that (1) For intersection queries, model with two layers can handle them well, more layers lead to overfitting. (2) For multi-hop queries, more layers are needed to model long dependencies in the query graphs.

| | #params | $A_s$ | $A_c$ | $A_p$ | $A_n$ |
|---|---|---|---|---|---|
| Q2T (NF) | 56M | 44.3 | 22.0 | 24.5 | 5.8 |
| Q2T (F) | 26M | **48.6** | **25.1** | **27.7** | **7.3** |

Table 3: Impact of whether freezing KGE on performance. Q2T (NF) denotes Not Freezing KGE, while Q2T (F) denotes Freezing KGE. $A_s$, $A_c$ denotes the average MRR of simple queries and complex queries. The number of parameters of the KGE is 30M.

## 5.4 Training with trainable KGE

Aiming to further prove the necessity of decoupling the training for simple queries and complex queries, on the FB15k dataset, we conduct experiments on Q2T with frozen/not frozen pretrained KGE. The results are shown in Table 3, where Q2T (NF) means training KGE and encoder on simple and complex queries together, Q2T (F) means only training encoder for complex queries. Although Q2T (NF) has more trainable parameters, it perform worse than Q2T (F). Especially on simple queries, a neural link predictor trained on simple queries can already handle it well, however, a more complex model trained on more complex queries leads to worse performance. As the final results are predicted by the pretrained link predictor (KGE), the worse the performance on simple queries, the worse the performance on complex queries.

## 6 Conclusion

In this paper, we present Q2T to transform diverse complex queries into a unified triple form that can be solved by pretrained neural link predictors. Q2T not only decouples the training of complex queries and simple queries from the training perspective, but also decouples KGE and the query encoder from the model perspective. The experiments on three datasets demonstrate that Q2T can achieve the state-of-the-art performance on almost all positive queries.

## 7 Limitations

The limitations of our proposed Q2T are as follows: (1) Q2T cannot handle negation queries well. Using more strategies to enable Q2T to answer negation queries well is a direction for future work. (2) We only use KGE as our pretrained link predictors, as the performance of Q2T is highly dependent on the pretrained link predictors. Other type models (e.g. GNN based model) as pretrained link predictors may perform better. (3) Limited to time and

resource, we only test performance of Q2T with different KGE models on the FB15k-237 queries.

## 8 Ethics Statement

This paper proposes a method for complex query answering in knowledge graph reasoning, and the experiments are conducted on public available datasets. As a result, there is no data privacy concern. Meanwhile, this paper does not involve human annotations, and there are no related ethical concerns.

## 9 Acknowledgment

This work was supported by National Key R&D Program of China (No.2022ZD0118501) and the National Natural Science Foundation of China (No.62376270, No.U1936207, No.61976211). This work was supported by the Strategic Priority Research Program of Chinese Academy of Sciences (No.XDA27020100), Youth Innovation Promotion Association CAS.

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

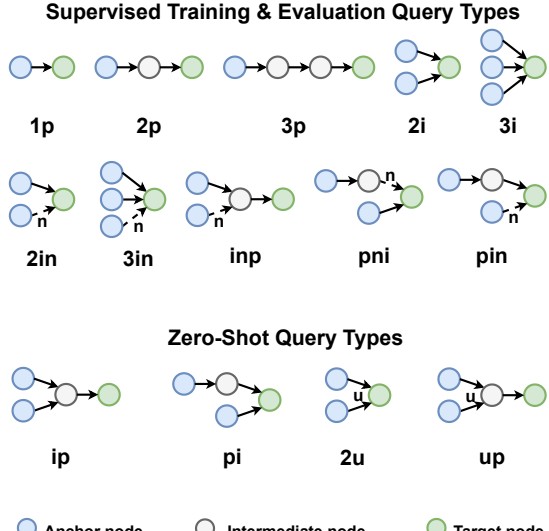

**Supervised Training & Evaluation Query Types**

1p    2p    3p    2i    3i

2in    3in    inp    pni    pin

**Zero-Shot Query Types**

ip    pi    2u    up

○ Anchor node    ○ Intermediate node    ● Target node

Figure 6: The illustration of all query types. The $p, i, n, u$ represent projection, intersection, negation, and union operations respectively.

## A  Data Details

The nine query types are shown in Figure 6. Specifically, there are five query types (1p/2p/3p/2i/3i) evaluated in a supervised manner, five query types (2in/3in/inp/pin/pni) evaluated in a few-shot manner, and four query types (2u/up/pi/ip) evaluated in a zero-shot manner. Given the query type, a sample is generated by random walking on the KG. The splitting of datasets are shown in Table 4. The average number of answers of test queries are shown in Table 5.

It should be noted that the training queries and the test queries are generated by random walking on $G_{train}$ (containing training edges) and $G_{test}$ (containing training, validation, and testing edges), respectively, which means that models need to predict at least one missing edges by knowledge graph reasoning.

## B  Experiment settings

For pretrained KGE model, we use the same settings as CQD (Arakelyan et al., 2021) and LMPNN (Wang et al., 2023b): ComplEx (Trouillon et al., 2016) model with rank 1000. The difference is that we use relation prediction as an auxiliary training objective in the training of ComplEx, as stated in section 4.1, so our ComplEx model perform better on 1p queries. The MRR scores of baselines we use are reported by Wang et al..

The default setting of Q2T is: 6 layer, 768 hidden

size, 12 number of heads. We tune the hyperparameters of Q2T on the validation set for each dataset by grid search. We consider the batch size from {512, 1024, 2048}, learning rate from {1e-4, 2e-4, 4e-4, 5e-4}, and label smoothing from {0.2, 0.4, 0.6, 0.8}. Our experiments are conducted on GTX 3090 with PyTorch 1.11, and the random seed are fixed for each experiment. The best hyperparameters for each datasets are shown in Table 6.

## C  Model parameters and training time

Table 7 show the number of parameters for models in Table 1. Although Q2T requires training KGE first and then training the query encoder, the overall training time is still shorter than that of other methods. This is due to the high training efficiency achieved by decoupling the training process of KGE and the query encoder.

BetaE (Ren and Leskovec, 2020), ConE (Zhang et al., 2021), Q2P (Bai et al., 2022) don't use pretrained KGE, they train neural set operators with KG embeddings from the zero, which is very inefficient. Although these methods seem to have less parameters than the pretrained KGE used in Q2T, they have much longer training time (e.g. The training times of BetaE and ConE are estimated to more than 20 hours under the official settings, while the training time for Q2T is less than 10 hours). Besides, the training for KGE on simple queries is efficient due to the simple structure of KGE, e.g., on the NELL queries, although ComplEx (Trouillon et al., 2016) model has 110M parameters, its training can be done in one hour.

## D  Label Smoothing

Aiming to prevent models from overfitting, label smoothing uses soft one-hop labels, instead of hard ones, to introduce noise during training. The smoothed label $y_{ls}$ is computed as follows:

$$y_{ls} = (1 - \alpha)y_h + \frac{\alpha}{K}$$

where $y_{ls}$ is the smoothed label, $y_h$ is the original label, $\alpha$ is the parameter of label smoothing, $K$ is the number of sampling. Lastly, standard cross-entropy loss is applied to these smoothed labels.

## E  Performance of Q2T with different pretrained KGE

Table 8 presents the MRR results of Q2T with different pretrained KGE on the FB15k-237 queries

| | Training | | Validation | | Test | |
|---|---|---|---|---|---|---|
| **Dataset** | 1p/2p/3p/2i/3i | others | 1p | others | 1p | others |
| FB15k | 273,710 | 27,371 | 59,097 | 8,000 | 67,016 | 8,000 |
| FB15k-237 | 149,689 | 14,968 | 20,101 | 5,000 | 22,812 | 5,000 |
| NELL | 107,982 | 10,798 | 16,927 | 4,000 | 17,034 | 4,000 |

Table 4: Number of training, validation, and test queries generated for different query types.

| Dataset | 1p | 2p | 3p | 2i | 3i | ip | pi | 2u | up | 2in | 3in | inp | pin | pni |
|---|---|---|---|---|---|---|---|---|---|---|---|---|---|---|
| FB15k | 1.7 | 19.6 | 24.4 | 8.0 | 5.2 | 18.3 | 12.5 | 18.9 | 23.8 | 15.9 | 14.6 | 19.8 | 21.6 | 16.9 |
| FB15k-237 | 1.7 | 17.3 | 24.3 | 6.9 | 4.5 | 17.7 | 10.4 | 19.6 | 24.3 | 16.3 | 13.4 | 19.5 | 21.7 | 18.2 |
| NELL995 | 1.6 | 14.9 | 17.5 | 5.7 | 6.0 | 17.4 | 11.9 | 14.9 | 19.0 | 12.9 | 11.1 | 12.9 | 16.0 | 13.0 |

Table 5: Average number of answers of test queries in datasets generated by Ren and Leskovec.

| Dataset | batchsize | dropout | ls | lr |
|---|---|---|---|---|
| FB15k | 1024 | 0.1 | 0.4 | 4e-4 |
| FB15k-237 | 1024 | 0.1 | 0.6 | 4e-4 |
| NELL | 1024 | 0.1 | 0.6 | 5e-4 |

Table 6: The best hyperparameters of Q2T on datasets generated by Ren and Leskovec. The *ls* and *lr* denote label smoothing and learning rate respectively.

generated by Ren and Leskovec. CP (Lacroix et al., 2018), DistMult (Yang et al., 2015) and ComplEx (Trouillon et al., 2016) are trained under the best hyperparameters settings provided by Chen et al., while Tucker (Balazevic et al., 2019) is trained under the best hyperparameters settings provided by Balazevic et al.. It can be observed that the performance of link predictors directly affects the final performance on complex queries. We also find that Q2T with different KGE models may be good at solving different queries.

## F Comparison with more baselinse on EPFO datasets

Table 10 shows the HIT@3 results of different CQA models on EPFO queries generated by Ren et al.. KgTransformer (Liu et al., 2022) utilizes two-stage pre-training : The first stage aims to initialize KgTransformer with KGs' general knowledge, and the second stage refines its ability for small queries during inference. Although KgTransformer has more parameters (8 layers, 1024 hidden size) and spends more time on more pre-trianing data, it still performs worse than our Q2T. The potential reason that KgTransformer perform better on pi/ip queries is that these type of queries may included in its pre-training data, which are randomly sampled from KGs.

## G Comparison with symbolic-integrated model

In contrast to non symbolic-integrated methods (e.g., BetaE (Ren and Leskovec, 2020), Q2P (Bai et al., 2022), CQD-CO (Arakelyan et al., 2021)), which use fixed-dimension intermediate embeddings (e.g., $10^2$), symbolic-integrated methods (e.g., GNN-QE (Zhu et al., 2022)) employ intermediate fuzzy set sizes that scale linearly with the Knowledge Graph size (e.g., $10^4$).

We select GNN-QE as a representative of symbolic-integrated methods for comparison with our Q2T, the results are shown in Table 12. It should be noticed that, GNN-QE employs NBFNet (Zhu et al., 2021) which applies a multi-layers GNN on the whole KGs for each projection operation. Besides, GNN-QE applies fuzzy logic operations to fuzzy sets of entities, so each node needs to obtain the probability list of all entities (a tensor with shape $(|V|,)$). As a result, GNN-QE requires much more training time and lots of GPU resources, it requires 128GB GPU memory to run a batch size of 32, while Q2T only demands 12GB GPU memory to run a batch size of 1024. Even then this, Q2T is also competitive with GNN-QE.

| Model | FB15k | | FB15k-237 | | NELL | |
|---|---|---|---|---|---|---|
| | #pretrained KGE params | #trainble params | #pretrained KGE params | #trainble params | #pretrained KGE params | #trainble params |
| BetaE | - | 19M | - | 20M | - | 58M |
| ConE | - | 28M | - | 24M | - | 63M |
| Q2P | - | 11M | - | 9.9M | - | 29M |
| (with pretrained KGE) | | | | | | |
| CQD | 35M | 0M | 30M | 0M | 128M | 0M |
| LMPNN | 35M | 16M | 30M | 16M | 128M | 16M |
| Q2T | 35M | 26M | 30M | 26M | 128M | 26M |

Table 7: The number of parameters for each model. CQD only use pretrained KGE to obtain answers with optimization-based method, so it doesn't have other trainable parameters.

| KGE Model | 1p | 2p | 3p | 2i | 3i | pi | ip | 2u | up | 2in | 3in | inp | pin | pni | $A_p$ | $A_n$ |
|---|---|---|---|---|---|---|---|---|---|---|---|---|---|---|---|---|
| TuckER | 45.3 | 12.5 | 10.1 | 28.2 | 39.8 | 21.8 | 15.8 | 16.9 | 10.0 | **7.0** | 8.9 | 7.6 | 4.0 | 5.0 | 22.2 | 6.5 |
| CP | 46.3 | 14.9 | 12.1 | 37.1 | 51.3 | **28.6** | 17.8 | 17.4 | 12.0 | 6.0 | 11.7 | **8.8** | **4.7** | 3.9 | 26.4 | 7.0 |
| DistMult | 47.2 | 15.1 | 11.9 | 37.2 | 50.9 | 28.1 | 18.3 | 18.7 | 12.2 | 6.2 | 12.5 | 8.4 | 4.5 | 4.1 | 26.6 | 7.1 |
| ComplEx | **48.5** | **16.2** | **12.6** | 37.8 | 51.8 | 28.6 | **19.2** | **21.9** | **12.5** | 6.4 | **12.7** | 8.5 | 4.6 | **4.2** | 27.7 | 7.3 |

Table 8: The MRR results of Q2T with different pretrained KGE on the FB15k-237 queries generated by Ren and Leskovec.

| Dataset | Model | 1p | 2p | 3p | 2i | 3i | ip | pi | 2u | up | Avg |
|---|---|---|---|---|---|---|---|---|---|---|---|
| FB15k-237 | GQE | 0.405 | 0.213 | 0.153 | 0.298 | 0.411 | 0.085 | 0.182 | 0.167 | 0.160 | 0.230 |
| | Q2B | 0.467 | 0.240 | 0.186 | 0.324 | 0.453 | 0.108 | 0.205 | 0.239 | 0.193 | 0.268 |
| | EmQL | 0.389 | 0.201 | 0.154 | 0.275 | 0.386 | 0.101 | 0.184 | 0.115 | 0.165 | 0.219 |
| | CQD ( CO ) | 0.512 | 0.213 | 0.131 | 0.352 | 0.457 | 0.146 | 0.222 | 0.281 | 0.132 | 0.272 |
| | CQD ( Beam ) | 0.512 | 0.288 | 0.221 | 0.352 | 0.457 | 0.129 | 0.249 | 0.284 | 0.121 | 0.290 |
| | KgTransformer | 0.459 | 0.312 | **0.276** | 0.398 | 0.528 | **0.189** | **0.286** | 0.263 | 0.214 | 0.325 |
| | Q2T | **0.530** | **0.329** | **0.276** | **0.415** | **0.529** | 0.167 | 0.251 | **0.358** | **0.230** | **0.343** |
| NELL | GQE | 0.417 | 0.231 | 0.203 | 0.318 | 0.454 | 0.081 | 0.188 | 0.200 | 0.139 | 0.248 |
| | Q2B | 0.555 | 0.266 | 0.233 | 0.343 | 0.480 | 0.132 | 0.212 | 0.369 | 0.163 | 0.306 |
| | EmQL | 0.456 | 0.231 | 0.172 | 0.331 | 0.483 | 0.143 | 0.244 | 0.226 | 0.207 | 0.277 |
| | CQD ( CO ) | 0.667 | 0.265 | 0.220 | **0.410** | 0.529 | 0.196 | 0.302 | 0.531 | 0.194 | 0.368 |
| | CQD ( Beam ) | 0.667 | 0.350 | 0.288 | **0.410** | 0.529 | 0.171 | 0.277 | 0.531 | 0.156 | 0.375 |
| | KgTransformer | 0.625 | 0.401 | 0.367 | 0.405 | **0.546** | **0.203** | **0.306** | 0.469 | 0.270 | 0.399 |
| | Q2T | **0.670** | **0.409** | **0.373** | 0.397 | 0.543 | 0.198 | 0.282 | **0.534** | **0.316** | **0.414** |

Table 10: HIT@3 results of different CQA models on EPFO queries generated by Ren et al.

| Dataset | Model | 1p | 2p | 3p | 2i | 3i | pi | ip | 2u | up | 2in | 3in | inp | pin | pni | $A_p$ | $A_n$ |
|---|---|---|---|---|---|---|---|---|---|---|---|---|---|---|---|---|---|
| FB15k-237 | GNN-QE | 42.8 | 14.7 | 11.8 | **38.3** | **54.1** | **31.1** | 18.9 | 16.2 | 13.4 | **10.0** | **16.8** | **9.3** | **7.2** | **7.8** | 26.8 | **10.2** |
| | Q2T | **48.4** | **15.6** | **12.4** | 37.8 | 51.9 | 28.6 | **19.4** | **21.8** | **12.8** | 6.1 | 12.4 | 8.4 | 4.5 | 4.1 | **27.6** | 7.1 |
| NELL | GNN-QE | 53.3 | 18.9 | 14.9 | **42.4** | **52.5** | **30.8** | 18.9 | 15.9 | 12.6 | **9.9** | **14.6** | **11.4** | **6.3** | **6.3** | 28.9 | **9.7** |
| | Q2T | **61.1** | **22.1** | **18.0** | 41.1 | 52.3 | 30.3 | **23.4** | **19.5** | **15.3** | 5.8 | 7.5 | 11.3 | 3.7 | 4.3 | **31.5** | 6.5 |

Table 12: Comparison between Q2T and GNN-QE.