# OpenReview forum: "Query2Triple: Unified Query Encoding for Answering Diverse Complex Queries over Knowledge Graphs"
_EMNLP/2023/Conference — EMNLP 2023 Findings_

### Official Review · Reviewer_EuCv · 2023-07-20

**Typos Grammar Style And Presentation Improvements:** In Table 7, the model is named CQC bu…
**Soundness:** 4

**Ethical Concerns:**

Yes

**Excitement:**

4: Strong: This paper deepens the understanding of some phenomenon or lowers the barriers to an existing research direction.

**Missing References:**

This paper could also discuss some of the baselines that appeared in EMNLP last year and ACL this year. A further literature review is suggested.

1. Dong Yang, Peijun Qing, Yang Li, Haonan Lu, and Xiaodong Lin. 2022. GammaE: Gamma Embeddings for Logical Queries on Knowledge Graphs. In Proceedings of the 2022 Conference on Empirical Methods in Natural Language Processing, pages 745–760, Abu Dhabi, United Arab Emirates. Association for Computational Linguistics.
2. Zihao Wang, Weizhi Fei, Hang Yin, Yangqiu Song, Ginny Wong, and Simon See. 2023. Wasserstein-Fisher-Rao Embedding: Logical Query Embeddings with Local Comparison and Global Transport. In Findings of the Association for Computational Linguistics: ACL 2023, pages 13679–13696, Toronto, Canada. Association for Computational Linguistics.

**Paper Topic And Main Contributions:**

This paper proposed a graph transformer-based approach (Query2Triple) for knowledge graph logical query answering, building upon the pretrained knowledge graph embeddings. Query2Triple proposed novel ways to use both the scoring function and pretrained embeddings of existing knowledge graph embedding approaches. The graph transformer model estimates virtual head and relation embeddings so the query answering problem is transformed into a tail estimation problem. It demonstrated strong performances on EPFO queries but suboptimal performances on negation queries.

**Questions For The Authors:**

1. What is the explicit form of $b_{\phi(v_i, v_j)}$? What does it mean by saying scalar indexed by $\phi(v_i, v_j)$.
2. How is label smoothing applied to the negative sampling loss.

**Reasons To Accept:**

1. Q2T proposed novel ways of using KG embeddings.
2. Q2T reaches state-of-the-art performance, particularly on EPFO queries. Though the scores on negation queries are not stronger than LMPNN, the overall performance is still competitive.

**Reasons To Reject:**

1. Some of the key technical details are missing. It is not clear how label smoothing is applied to the negative sampling loss. The coherence should be improved. For example, in Table 7, the model is named CQC but not Q2T as in the rest of this paper.
2. This paper seems to overclaim its contribution. As an important baseline, LMPNN also proposes to only train neural networks and keep the KG embeddings unchanged. In the LMPNN paper, it states that "Extensive experiments show that our approach is a new state-of-the-art neural CQA model, in which only one MLP network and two embedding vectors are trained." Therefore, it is questionable to argue that

>"Compared to LMPNN, Q2T further decouples the training for simple and complex queries." in lines 206-208

>"Q2T (NF) means training KGE and encoder on simple and complex queries together (the same strategy as LMPNN)" in lines 542-545,

so as the comparisons in Table 7. I suggest that the authors double-check this carefully.

**Reproducibility:**

5: Could easily reproduce the results.

**Reviewer Confidence:**

5: Positive that my evaluation is correct. I read the paper very carefully and I am very familiar with related work.

---

> ### Author Rebuttal · Authors · 2023-08-28
>
> Dear reviewer EuCv,
>
> Thanks for your informative suggestions. Here is our feedback.
>
>
>
> > Q1: Some of the key technical details are missing. It is not clear how label smoothing is applied to the negative sampling loss.
>
> A1: Due to page limitations, we only represent our original loss function in formula 14, label smoothing utilizes soft one-hop labels instead of hard ones to add noise in the training, reduces the weights of true labels when calculating the training loss and thus prevents overfitting when training, the label after smoothing is as follows:
> $$
> y_{ls} = (1-\alpha) y_h+\frac{\alpha}{K}
> $$
> where $y_{ls}$ is the smoothed label, $y_h$ is the original label, $\alpha$ is the parameter of label smoothing, $K$ is the number of sampling. Lastly, standard cross-entropy loss is applied to these smoothed labels.
>
> > The coherence should be improved. For example, in Table 7, the model is named CQC but not Q2T as in the rest of this paper.
>
> It should be Q2T, we apologize for forgetting to rename it in the appendix.
>
>
>
> > Q2: As an important baseline, LMPNN also proposes to only train neural networks and keep the KG embeddings unchanged. In the LMPNN paper, it states that "Extensive experiments show that our approach is a new state-of-the-art neural CQA model, in which only one MLP network and two embedding vectors are trained."
>
> A2: We found that the official repository of LMPNN was updated once after we submitted the paper. In the new code version, LMPNN also maintains KGE unchanged during subsequent training. The updated comparison between Q2T and LMPNN is as follows:
>
> 1.  In LMPNN, the $\rho$ function of KGE is invoked during each message propagation for each edge, whereas Q2T only requires a single invocation of the KGE score function for any type of complex query. Therefore, Q2T is better suited for leveraging KGE with higher complexity and larger parameters.
> 2. In LMPNN, the coupling of KGE and GNN necessitates the design of distinct logical message propagation functions (predicting head entity representations, predicting tail entity representations) for different types of score functions in KGE, as well as for different types of regularization optimization. However, Q2T can be employed with any KGE without the need for additional code modifications. Therefore, Q2T is more modular.
>
>
>
> > Q3: What is the explicit form of $b_{\phi(v_i,v_j)}$, What does it mean by saying scalar indexed by $\phi(v_i,v_j)$.
>
> A3: $b_i$ means the $t$-th element of vector $\boldsymbol{b}$, which is a scalar. When $i=\phi(v_i,v_j)$, $\phi(v_i,v_j)$ is the index.
>
>
>
> > Q4: For Missing References of GammaE and Wasserstein-Fisher-Rao Embedding
>
> A4: We have noticed these works, due to page limitations and Wasserstein-Fisher-Rao Embedding was not published by ACL 2023 when we submitted this article, we didn't add them to Table1 although Q2T outperforms them, we will add them in the updated version.
>
>
>
> Best regards,
>
> Authors of Paper1769

---

### Official Review · Reviewer_hmQo · 2023-08-04

**Typos Grammar Style And Presentation Improvements:** 1. Typo
**Soundness:** 4

**Excitement:**

4: Strong: This paper deepens the understanding of some phenomenon or lowers the barriers to an existing research direction.

**Missing References:**

[1] Xuelu Chen, Ziniu Hu, and Yizhou Sun. Fuzzy logic-based logical query answering on knowledge graphs. In Thirty-Sixth AAAI Conference on Artificial Intelligence, AAAI 2022

[2] Jiaxin Bai, Tianshi Zheng, and Yangqiu Song. Sequential query encoding for complex query answering on knowledge graphs. Transactions on Machine Learning Research, 2023


**Paper Topic And Main Contributions:**

This paper proposes to embed the linearized query graph with KG embedding, then use a refined transformer-based model to encode it to head-relation representations. It achieved state-of-the-art results on three datasets in EPFO queries, with a relatively small number of parameters.


**Questions For The Authors:**



1. Missing baseline: FuzzQE [1] is a strong query encoding baseline without kg-embedding, it’s encouraged to add it into comparison.

2. A contemporary work SQE [2] is highly related to this paper, it’s encouraged to add it to the discussion.


**Reasons To Accept:**

1. As stated in “main contributions”, The paper introduces a novel perspective toward query encoding.

2. The paper included the result from different KGE models, which substantiate the effectiveness of KG embedding in the whole CQA system.

3. Also, The discussion on whether freezing the pre-trained KGE is quite insightful.


**Reasons To Reject:**

1. Lack of ablation study. The major proposed design (also the main difference between Q2T and KgTransformer) is the answer decoding approach. KgTransformer outputs the final node embedding directly, while Q2T outputs two vectors (head, relation) and then decodes with a link predictor. However, the paper didn’t provide an ablation study to prove the importance of this design. Is the improvement brought by the transformer backbone or the link predictor?

2. As stated in the paper, the Q2T model does not outperform the previous state-of-the-art models on negative queries, despite its significant improvement on positive queries. This suggests the poor ability of Q2T to handle the "complement" operator. It is doubtful whether Q2T can maintain its performance when generalizing to more unseen query types.




**Reproducibility:**

4: Could mostly reproduce the results, but there may be some variation because of sample variance or minor variations in their interpretation of the protocol or method.

**Reviewer Confidence:**

5: Positive that my evaluation is correct. I read the paper very carefully and I am very familiar with related work.

---

> ### Author Rebuttal · Authors · 2023-08-28
>
> Dear reviewer hmQo,
>
> Thanks for your informative suggestions. Here is our feedback.
>
>
>
> > Q1: Lack of ablation study. The major proposed design (also the main difference between Q2T and KgTransformer) is the answer decoding approach. KgTransformer outputs the final node embedding directly, while Q2T outputs two vectors (head, relation) and then decodes with a link predictor. However, the paper didn’t provide an ablation study to prove the importance of this design. Is the improvement brought by the transformer backbone or the link predictor?
>
> A1: We have conducted the experiment that makes Q2T output the final node embedding directly in the early time, but the performance is extremely poor. We perceive that only if Transform-based models have more parameters and larger training data, can they fully record the structure of KGs, and this is what KgTransformer does.
>
> Specifically, on the FB15k-237 dataset, **KgTransformer has more than 600M trainable parameters**, while Q2T has only 56M trainable parameters (30M for KGE, 26M for query encoder). Besides, KgTransformer uses two-stage pre-training (Initialization and Refinement) and then fine-tuning, which not only requires much more training time than Q2T but also needs to sample more data by random walking on KGs.
>
> Both Q2T and KgTransformer use the transformer as backbone, but Q2T can outperform KgTransformer easily with much less parameters and training time. Therefore, we believe this improvement is brought by our decoding approach which only uses the transformer to generate embeddings for retrieving answers from KGE.
>
> In the later version,  we will compare Q2T and KgTransformer in more detail.
>
>
>
> > Q2: As stated in the paper, the Q2T model does not outperform the previous state-of-the-art models on negative queries, despite its significant improvement on positive queries. This suggests the poor ability of Q2T to handle the "complement" operator. It is doubtful whether Q2T can maintain its performance when generalizing to more unseen query types.
>
> A2: We perceive the poor ability of Q2T to handle the "complement" operator is due to the KGE that is only trained on positive factual triples (line 488-492), because the distribution of answer entities of queries without negation operators is more concentrated, while the distribution of answer entities of queries with negation operators is more dispersed (as the "complement" operator may lead lots of candidate answers, such as "Who is not the winner of the Turning Award").
>
> We also computed the average 2-norm distance of candidate answer entity embeddings sets across different query types, denoted as $cdist$ (the higher the value of cdist, the more dispersed the distribution of embeddings within this set). Then we found that the $cdist$ of positive queries is smaller than its corresponding negative queries consistently, e.g., $cdist(ip)=9.08$, while $cdist(inp)=9.55$,  $cdist(2i)=7.73$, while $cdist(2in)=8.88$. The pretrained KGE models struggle to generate appropriate embeddings for queries with dispersed answer distributions, as they are only trained on positive factual triples with concentrated answer distributions. In the updated version, we will discuss this question in more detail
>
> Besides, Q2T also performs well on ip/pi queries, which are tested in zero-shot learning (described in Table 4). This means Q2T can generalize well on unseen query types without  "complement" operator.
>
>
>
> > Q3: For Missing References of FuzzQE and SQE
>
> A3: We have noticed these works,  due to page limitations, we didn't add them to Table1 although Q2T outperforms them, we will add them in the updated version.
>
>
>
> > Q4: Typo: what is “CQC” in table 7?
>
> A4: It should be Q2T, we apologize for forgetting to rename it in the appendix.
>
>
>
> Best regards,
>
> Authors of Paper1769

---

### Official Review · Reviewer_Tnh9 · 2023-08-05

**Soundness:** 2

**Excitement:**

3: Ambivalent: It has merits (e.g., it reports state-of-the-art results, the idea is nice), but there are key weaknesses (e.g., it describes incremental work), and it can significantly benefit from another round of revision. However, I won't object to accepting it if my co-reviewers champion it.

**Missing References:**

GNN-QE: Graph Neural Network Query Executor

@inproceedings{zhu2022neural,
  title={Neural-symbolic models for logical queries on knowledge graphs},
  author={Zhu, Zhaocheng and Galkin, Mikhail and Zhang, Zuobai and Tang, Jian},
  booktitle={International Conference on Machine Learning},
  pages={27454--27478},
  year={2022},
  organization={PMLR}
}

**Paper Topic And Main Contributions:**

The paper present a novel method for complex query answering in Knowledge Graphs (KG). The approach is to decouple the training on simple one-hop (link-prediction) queries and complex queries into modular parts. The authors suggest the use of a complex query encoder to be used in conjunction with a pre-trained link predictior.

**Reasons To Accept:**

1) The authors suggest a novel method for tackling complex query answering in knowledge graphs.
2) The approach allows for faster inference and tackles the computationally expansive nature of query answering.
3) The ablation w.r.t. label smoothing used in the method is solid.

**Reasons To Reject:**

1) Methods such as GNN-QE also train on link-prediction (1p) queries and tune with Complex queries with none/minor performance degradation on simpler queries occurring. This directly contradicts the claim in the abstract (lines 007-013).

2) (lines 068-071) The discrepancy that models cannot perform well on both link-prediction (1p) and more complex queries is not justified at all and should be further validated/elaborated.

3) (lines 092-093) QE models are not the only ones suffering from error cascading. Methods such as CQD also compute intermediate candidates and are prone to error cascading.

4) (lines 188-196) CQD does not handle negation at all. The original study does not include results with negations; thus, comparing CQD with negations is unjustified. Please elaborate on why the method is "time-consuming"?

5) The numbers in Table 1 are inconsistent with the original results:
@inproceedings{minervini2022complex,
  title={Complex query answering with neural link predictors},
  author={Minervini, Pasquale and Arakelyan, Erik and Daza, Daniel and Cochez, Michael}
}

6) The current SOTA GNN-QE is completely omitted from the baselines, although it is the closest work in terms of components to this approach. This baseline should be added and compared.

7) It is very peculiar and dubious to see that a method for complex query answering not only mitigates the effect of training for simple and complex query answering but also overperforms models that are particularly trained for link prediction on all the benchmarks.

8) Does the method include an interoperability component, as a portion of complex query-answering methods do? Most methods can assess error cascading through these means.

9) The contribution seems incremental compared to existing methods such as GNN-QE, CQD and BetaE.

**Reproducibility:**

2: Would be hard pressed to reproduce the results. The contribution depends on data that are simply not available outside the author's institution or consortium; not enough details are provided.

**Reviewer Confidence:**

5: Positive that my evaluation is correct. I read the paper very carefully and I am very familiar with related work.

---

> ### Author Rebuttal · Authors · 2023-08-28
>
> Dear reviewer Tnh9,
>
> Thanks for your informative suggestions. Here is our feedback.
>
> > Q1: GNN-QE is trained on link-prediction (1p) queries and tuned with Complex queries with none/minor performance degradation on simpler queries occurring, which contradicts the claim in the abstract.
>
> A1: In this work, we aim to compare our work with Query Embedding (QE) based methods, such as BetaE, Q2P and so on. For these methods, the intermediate embeddings are of fixed dimensions. However, Graph Neural Network Query Executor (GNN-QE) belongs to symbolic integrated methods, For these methods, the sizes of intermediate fuzzy sets grow linearly with the size of the Knowledge Graph, which means they require significantly more GPU resources and do not scale well for large Knowledge Graphs. The claim in the abstract (line 003-013) refers to Query Embedding based methods, and only QE based methods include neural set operators (line 009).
>
> Besides, GNN-QE is not trained on link-prediction (1p) queries and tuned with Complex queries, it is trained on 10 query types concurrently (1p/2p/3p/2i/3i/2in/3in/inp/pni/pin)  without pretrained embedding models (as stated in section 2 and section 5.1 of  the GNN-QE original paper).
>
> Although GNN-QE utilizes more GPU resources and more training time to achieve higher performance on complex queries, simple Knowledge Graph Embedding (KGE) models (such as the ComplEx model used in CQD) can still outperform it easily on 1p queries (as shown in Table 1 of the GNN-QE original paper).
>
> In summary, our claim in abstract refers to Query Embedding based methods, and GNN-QE,  as a symbolic integrated method, also confirms our claim.
>
>
>
> > Q2: The discrepancy that models cannot perform well on both link-prediction (1p) and more complex queries is not justified at all and should be further validated/elaborated
>
> A2: From an experimental perspective, we can find  from Table 1 that even QE based models such as BetaE, Q2P and ConE model the projection operator with complex neural networks, traditional Knowledge Graph Embedding (KGE) models, such as ComplEx model used by CQD and Q2T, can still easily outperform them on link prediction (1p) (line 071-076).
>
> From a theoretical perspective, there are two reasons:
>
> 1. The complex neural networks employed in these studies for modeling the projection operator are incapable of effectively managing inherent relational properties, such as symmetry, asymmetry, inversion, composition, and more, which are sufficiently studied in KG completion tasks and addressed by KGE. However most QE methods don't make full use of the well-studied KGE.
>
> 2. The training for simple queries and complex queries are coupled. Therefore, the entity and relation embeddings inevitably incorporate neural set operators information that is irrelevant for answering link-prediction queries (1p), leading to a decrease in performance (line 076-083).
>
> To solve these drawbacks we use a competitive KGE model (ComplEx-N3-RP, ComplEx with N3 Regularizer and relation prediction, line 258-274) to handle link prediction, and combine it with an extra encoder to solve complex queries.
>
> The experiment in section 5.2 (line 537-555) further validates our assumption as it proves the necessity of decoupling the training for simple queries and complex queries. Specifically,  if we do not freeze the KGE and tune the pretrained ComplEx model and the query encoder on link prediction (1p) queries and complex queries together, we get worse performance even introducing more parameters.
>
> We will discuss this question in more detail in subsequent versions.
>
>
>
> > Q3: QE models are not the only ones suffering from error cascading. Methods such as CQD also compute intermediate candidates and are prone to error cascading.
>
> A3: Yes, CQD is prone to error cascading too, but we focus mainly on the general drawbacks of Query Embedding based methods in our paper. Due to page limitations, we only mention the two main disadvantages of CQD in the later part of the article (line 188-196), we will discuss the disadvantages of CQD in more detail in subsequent versions.
>
>
>
> > Q4: CQD does not handle negation at all. The original study does not include results with negations; thus, comparing CQD with negations is unjustified.
> >
> > Q5: The numbers of CQD in Table 1 are inconsistent with the original results.
>
> A4: As our work follows LMPNN, the baseline results in Table 1 are taken from the original paper of LMPNN. LMPNN extended CQD to negation queries with the continuous truth value with fuzzy logical negator and tested it on the dataset introduced by BetaE, while the original CQD is tested on the dataset introduced by Query2Box (doesn't contain negation queries). So the performance of CQD in Table 1 is different with the original results.
>
> In order to clear up the misunderstanding, we will rename CQD to CQD (E)  in Table 1 in subsequent versions.
>
> > Please elaborate on why the method is "time-consuming"
>
> Due to page limitations, we didn't elaborate how CQD works. CQD utilizes the KG representation to calculate the continuous truth value of an EPFO logical formula with the logical t-norms. Then, the embeddings are optimized to maximize the continuous truth value. When the optimization is applied in the embedding space,  gradient-based optimisation methods (such as Adam and SGD) are used to update node embeddings multiple times until convergence or reaching the maximum number of iterations, which is time-consuming.
>
> When the optimization is applied in the symbolic space, it uses beam search, which requires more time for answering queries (Figure 4 of their original paper).
>
> We will discuss more details of CQD in subsequent versions.
>
>
>
> > Q6:The current SOTA GNN-QE is completely omitted from the baselines, although it is the closest work in terms of components to this approach.
>
> A6: Firstly, LMPNN (ICLR 2023) is  closer to our work than GNN-QE (ICML 2022). Secondly, as stated in A1, in this work, we aim to compare our work with Query Embedding (QE) based methods with intermediate embeddings of fixed dimension (10^2). GNN-QE is a symbolic integrated method where intermediate fuzzy set sizes scale linearly with the Knowledge Graph size (10^4). Besides, GNN-QE employs NBFNet (Zhu et al., 2021) which applies GNN on the whole KG (training set) for each message passing iteration. So GNN-QE requires much more training time and lots of GPU resources (4 V100 GPU (32G), which is 10 times larger than the resources required by Q2T) . This means that GNN-QE suffers from the scalabilities issues.
>
> Despite this, Q2T can still outperform GNN-QE on EPFO queries over FB15k-237 and NELL, the results are as follows.
>
> |  **Dataset**  | **Model** |  **1p**   |  **2p**   |  **3p**   |  **2i**   |  **3i**   |  **pi**   |  **ip**   |  **2u**   |  **up**   |  **2in**  |  **3in**  |  **inp**  | **pin**  | **pni**  | $\boldsymbol{A_p}$ | $\boldsymbol{A_n}$ |
> | :-----------: | :-------- | :-------: | :-------: | :-------: | :-------: | :-------: | :-------: | :-------: | :-------: | :-------: | :-------: | :-------: | :-------: | :------: | :------: | :----------------: | :----------------: |
> | **FB15k-237** | GNN-QE    |   42\.8   |   14\.7   |   11\.8   | **38\.3** | **54\.1** | **31\.1** |   18\.9   |   16\.2   |   13\.4   | **10\.0** | **16\.8** | **9\.3**  | **7\.2** | **7\.8** |       26\.8        |     **10\.2**      |
> |               | Q2T       | **48\.4** | **15\.6** | **12\.4** |   37\.8   |   51\.9   |   28\.6   | **19\.4** | **21\.8** | **12\.8** |   6\.1    |   12\.4   |   8\.4    |   4\.5   |   4\.1   |     **27\.6**      |        7\.1        |
> |   **NELL**    | GNN-QE    |   53\.3   |   18\.9   |   14\.9   | **42\.4** | **52\.5** | **30\.8** |   18\.9   |   15\.9   |   12\.6   | **9\.9**  | **14\.6** | **11\.4** | **6\.3** | **6\.3** |       28\.9        |      **9\.7**      |
> |               | Q2T       | **61\.1** | **22\.1** | **18\.0** |   41\.1   |   52\.3   |   30\.3   | **23\.4** | **19\.5** | **15\.3** |   5\.8    |   7\.5    |   11\.3   |   3\.7   |   4\.3   |     **31\.5**      |        6\.5        |
>
> Regarding the FB15k dataset, the performance of GNN-QE is exceptionally strong. We argue that this phenomenon stems from data leakage (stated in the orignal paper of FB15k-237), where a large number of test triples could be obtained by inverting triples in the training set. Consequently, GNN-QE might exploit these shortcuts while applying GNN on the whole KG (training set) to get answers.
>
> In the later version, we will compare our Q2T and other symbolic integrated methods including GNN-QE in the appendix.
>
>
>
> > Q7: It is very peculiar and dubious to see that a method for complex query answering not only mitigates the effect of training for simple and complex query answering but also overperforms models that are particularly trained for link prediction on all the benchmarks.
>
> A7: The key idea of our work is: (1) First, train a KGE model on link prediction queries. (2) Then, train a query encoder for complex queries (the pretrained KGE is frozen) (line 098-126).
>
> For link prediction queries, we just use the pretrained KGE model to solve them. For complex queries, we use the query encoder to transform them into a unified triple form which can be solved by the pretrained KGE model (Figure 3 (A)).
>
> The KGE model and the query encoder of Q2T are decoupled (line 130-135). Therefore, we can make full use of the existing mature technology in the field of KGE to train our KGE model. In this work, we use ComplEx-N3-RP (ComplEx with N3 Regularizer and relation prediction, line 258-274), which is a very competitive model for link prediction, as our KGE model, so Q2T can perform well on link prediction.
>
> Besides, the test 1p queries generated by BetaE are slightly different from the standard KG test datasets, so the performance on 1p can not be compared directly with that of the standard KGE models that are tested on the standard KG test datasets.
>
>
>
> > Q8: Does the method include an interoperability component, as a portion of complex query-answering methods do? Most methods can assess error cascading through these means.
>
> A8: Q2T uses an end-to-end reasoning mode, which means Q2T doesn't need to obtain the intermediate embedding of each variable node (Figure 4 and Figure 3 (A)). So, there is no error cascading in Q2T, and that's the reason why Q2T can outperform other models on path queries (2p/3p) (line 457-461).
>
>
>
> > Q9: The contribution seems incremental compared to existing methods such as GNN-QE, CQD and BetaE.
>
> A9: Q2T is the first work to treat CQA as a special link prediction task where it learns how to generate appropriate inputs for the pretrained neural link predictor end-to-end (line 122-126), which is completely different from the existing QE-based methods (BetaE, Q2P),  symbolic integrated methods (GNN-QE) and CQD.
>
> The results of this experiment demonstrate that Q2T, despite having fewer parameters and requiring less training time, can still outperform other baseline models. Furthermore, Q2T does not necessitate explicit modeling of neural set operators, making it a more concise model.
>
> What's more, Q2T further bridges the studies of KGE and CQA, as we are the first to prove that KGE models can be also used to answer complex logical queries directly without updating themselves. Specifically, each pretrained neural link predictor can be regarded as a neural knowledge base that record the structure of KG, while learning to transform complex queries into suitable inputs for the neural link predictor can be seen as an end-to-end process of retrieving answers from this neural knowledge base (line 471-477).
>
>
>
> Best regards,
>
> Authors of Paper1769

---

### Meta-Review · Area_Chair_oycX · 2023-09-17

**Recommendation:** 4

**Metareview:**

This paper proposed a graph transformer-based approach (Query2Triple) for knowledge graph logical query answering, building upon the pretrained knowledge graph embeddings.

Reasons to accept:
- The task addressed in the paper is relevant for EMNLP audience and it is supported by robust experiments and results.
- The paper proposes an interesting approach whose results are very competitive with SOTA.
- Analysis and discussion are robust and insightful for future research.

Reasons to reject:
- some concepts (baselines, contributions, and other minors, see reviews) can be described better in the text.

---

### Decision · Program_Chairs · 2023-10-07

**Decision:**

Accept-Findings

**Comment:**

This paper proposed a graph transformer-based approach (Query2Triple) for knowledge graph logical query answering, building upon the pretrained knowledge graph embeddings.

Reasons to accept:
- The task addressed in the paper is relevant for EMNLP audience and it is supported by robust experiments and results.
- The paper proposes an interesting approach whose results are very competitive with SOTA.
- Analysis and discussion are robust and insightful for future research.

Reasons to reject:
- some concepts (baselines, contributions, and other minors, see reviews) can be described better in the text.